# Estimates of scCO₂ Storage and Sealing Capacity of the Janggi Basin in Korea Based on Laboratory Scale Experiments

**Jinyoung Park [1,2], Minjune Yang [1] , Seyoon Kim [1], Minhee Lee [1,*] and Sookyun Wang [3]**

1   Department of Earth Environmental Sciences, Pukyong National University, Busan 48513, Korea
2   BK21 Plus Project of the Graduate School of Earth Environmental Hazard System,
    Pukyong National University, Busan 48513, Korea
3   Department of Energy Resources Engineering, Pukyong National University, Busan 48513, Korea
*   Correspondence: heelee@pknu.ac.kr; Tel.: +82-51-629-6630

**Abstract:** Laboratory experiments were performed to measure the supercritical CO₂ (scCO₂) storage ratio (%) of conglomerate and sandstone in the Janggi Basin, which are classified as rock in Korea that are available for CO₂ storage. The scCO₂ storage capacity was evaluated by direct measurement of the amount of scCO₂ replacing the pore water in each reservoir rock core. The scCO₂ sealing capacity of the cap rock (i.e., tuff and mudstone) was also compared by measuring the scCO₂ capillary entry pressure (Δp) into the rock core. The measured average scCO₂ storage ratio of the conglomerate and the sandstone were 30.7% and 13.1%, respectively, suggesting that the scCO₂ storage capacity was greater than 360,000 metric tons. The scCO₂ capillary entry pressure for the tuff ranged from 15 to 20 bar and for the mudstone it was higher than 150 bar, suggesting that the mudstone layers had enough sealing capacity from the aspect of hydromechanics. From XRF analyses, before and after 90 d of the scCO₂-water-cap rock reaction, the mudstone and the tuff were investigated to assure their geochemical stability as the cap rock. From the study, the Janggi Basin was considered an optimal CO₂ storage site based on both its high scCO₂ storage ratio and high capillary entry pressure.

**Keywords:** CO₂ reservoir rock; CO₂ sealing capacity; CO₂ sequestration; CO₂ storage capacity; CO₂ storage ratio; supercritical CO₂

## 1. Introduction

Eco-friendly plans and policies to reduce CO₂ emission are being driven forward around the world. In developed countries, CO₂ capture and sequestration (CCS) technology is partially commercialized and the total amount of subsurface CO₂ storage has been on the rise [1–4]. Since the early 2000s, the government of South Korea has been working on several projects to determine the optimal CO₂ storage sites on the Korean peninsula and the Janggi Basin. This basin is located in the southeastern part of the East Sea and is currently being evaluated for its potential to be one of the best onshore or offshore storage sites in Korea [5,6]. From geophysical and geological surveys, it is has become clear that the Miocene Janggi Basin consists of four small blocks (Guryongpo, Ocheon, Noeseongsan, and Yeongamri basins) [7]. The Noeseongsan block contains rudaceous sandstone and conglomerate layers that are considered promising for CO₂ storage sites more than 800 m deep; these also have mudstone and dacitic tuff layers above them that are able to serve as stable shield layers [8,9]. The Korean government has a plan to inject a hundred thousand metric tons of CO₂ in a pilot-scale onshore CO₂ storage test site in 2030 and the Janggi Basin is considered a suitable place for the scCO₂ storage test site. During stratigraphic analysis in 2015 and 2016, four sites in the Janggi Basin were drilled by the

Korea Institute of Geoscience and Mineral Resources (KIGAM), and continuous drill cores to 1200 m in depth were collected at each site. From previously collected well logging data and the geo-structural interpretation results, the Janggi conglomerate formation in the Janggi Basin can be divided into four lithofacies. These consist of conglomerate and rudaceous sandstone from gravelly braided stream deposits, coarse sandstone deposited in mouth-bar or delta, muddy sandstone and shale deposited in floodplain environments, and mudstone as lacustrine deposits [8]. The western part of the basin is mainly composed of thick conglomerate and rudaceous sandstone lithofacies, which are available for use as a $CO_2$ storage reservoir, whereas the mudstone and muddy sandstone lithofacies constitute the eastern part of the basin. The western part of the Janggi Basin has no large faults and is considered to be an optimal $CO_2$ storage area. This area is now estimated to be at least about 10 $km^2$, assuming that the practical volume of the $CO_2$ storage volume in the Janggi Basin deeper than 800 m is about 0.025 $km^3$ [8,9].

The estimation of the $CO_2$ storage capacity of geological storage reservoirs is essential to determine reasonable $CO_2$ storage site candidates and is directly dependent on the practical amount of $scCO_2$ in non-aqueous phase, which can be stored in the pore spaces of the reservoir rock after $scCO_2$ injection. The more water displaced by $scCO_2$ in the void spaces of the reservoir rock, the more $CO_2$ could be stored there. There are no exact definitions for $scCO_2$ storage capacity yet, although there are several definitions of storage capacity from previous studies [10–18]. In recent research, the $scCO_2$ storage capacity was generally defined as the proportion of the volume of $scCO_2$ stored after injection, in relation to the pore volume of the $CO_2$ reservoir rock [19,20]. Geological exploration and numerical simulation to determine the representative amount of $scCO_2$ that could be stored within the pore spaces of a specific reservoir formation have been major subjects in $CO_2$ sequestration (CS) studies.

However, the pore volume saturated with water was not fully replaced by $scCO_2$ while injecting $scCO_2$ into the reservoir rock and the practical $scCO_2$ storage capacity for the specific reservoir rock can often be overestimated. The $scCO_2$ displacement of water from pore spaces of the rock during $scCO_2$ injection can be influenced by various parameters—Not just physical properties such as pore size, heterogeneity of the pore network, and injection pressure, but also mineralogical and geochemical reactivity. Thus, the best way to determine the amount of $scCO_2$ storage possible in a specific rock formation is direct measurement of $scCO_2$ displacement of water under $scCO_2$ injection conditions. This occurs using rock cores at a laboratory scale, from which the results are extended to macro scale, including the entire reservoir formation [21,22].

The $scCO_2$ storage ratio (%) (or "$scCO_2$ displacement efficiency (%)") of a reservoir formation is one of the most general parameters used in laboratory work for evaluating the $CO_2$ storage amount of a formation [3,10,19,20]. The $scCO_2$ storage ratio is defined as the fraction of the amount of $scCO_2$ occupying pore spaces after $scCO_2$ injection into a reservoir formation. It can be directly measured under $scCO_2$ injection conditions simulated on a laboratory scale. In 2016, Wang et al. and Kim et al. [19,20] carried out direct laboratory measurement of the $scCO_2$ storage ratio for possible reservoir rock cores, presenting the possibility that it might be used to estimate the $scCO_2$ capacity for the specific reservoir. However, this process was still in the experimental stages and the number of studies on $scCO_2$ storage capacity based on direct measurement in the laboratory are very limited. The $CO_2$ storage capacity for a specific reservoir formation can be calculated by multiplying the $scCO_2$ storage ratio by the total void volume of the formation, ignoring the amount of dissolved $CO_2$. There are many benefits of direct measurement of the $scCO_2$ storage ratio because it represents the substantive amount of $CO_2$ retained in a specific storage formation, under the $scCO_2$ injection condition. The estimation of the practical amount of $CO_2$ storage for a specific reservoir rock could be made possible by using both the $scCO_2$ storage ratio and the reservoir volume acquired from geological survey, which has almost never been tried before. In this study, laboratory experiments were performed to measure the amount of $scCO_2$ displacing water from the pore spaces of the sandstones and conglomerate cores sampled from 800–1000 m depth in the Janggi Basin, which is classified as an available $CO_2$ storage reservoir in Korea. The $scCO_2$ storage capacity of the Janggi Basin was calculated

quantitatively, according to the measured $scCO_2$ storage ratio and with additional geophysical data on the spatial domain of the Janggi Basin.

The $scCO_2$ sealing capacity of the cap rock is another major parameter used to select successful $CO_2$ storage sites because it correlates to the leakage of $scCO_2$ during the anticipated duration of $CO_2$ sequestration. Even if a $CO_2$ storage site has enough $scCO_2$ storage capacity, it has to be excluded from suitable $CO_2$ storage sites if the $scCO_2$ leakage safety of the cap rock in the site is not assured. Layers of mudstone and dacitic tuff are repeated above the rudaceous sandstone and conglomerate layers in the Janggi Basin and it is assumed that they can play the role of cap rock to prevent the upward movement of $scCO_2$ from deeper reservoir rock [9]. In 2017, the initial capillary entry pressure of $scCO_2$ into the cap rock core surface, determined when the $scCO_2$ began to infiltrate the rock, was successfully measured in the laboratory [23]. In this study, several experimental conditions for the capillary entry pressure measurement such as the boundary condition of the high-pressure tank and the $scCO_2$ injection time on the core surface were modified, to more realistically simulate the $scCO_2$ leakage at the boundary between the reservoir and the cap rock. More information for the comparison of the experimental conditions can be drawn from [23]. The $scCO_2$ sealing capacity of the mudstone and dacitic tuff in the Janggi Basin was evaluated based on their initial $scCO_2$ capillary entry pressures under simulated $scCO_2$ injection P-T conditions. The change of mineralogical composition of the reservoir and cap rock after 90 d of $scCO_2$–water–rock reaction at 50 °C and 100 bar was also investigated by XRF analysis. This was done to observe the effect of $scCO_2$-related geochemical reactions on the sealing capacity. From the experimental results on the $scCO_2$ storage ratio for the reservoir rock, on the initial $scCO_2$ capillary entry pressure for the cap rock, and from mineralogical analyses for rock cores, the feasibility of the Janggi Basin as an available pilot-scale $CO_2$ storage test site where a hundred thousand metric tons of $CO_2$ could be injected was evaluated.

This study presents a novel and reliable method by which to select a successful $CO_2$ storage site based on both quantitative $scCO_2$ storage ratio and capillary entry pressure under $CO_2$ sequestration conditions, as well as on geochemical analyses. The results of this study will also provide ideas for further quantitative research about the $CO_2$ storage capacity and $CO_2$ leakage safety based on practical measurements of the $scCO_2$ storage ratio and initial $scCO_2$ capillary entry pressure.

## 2. Materials and Methods

### 2.1. Preparation of the scCO_2 Reservoir and Capping Rock Cores

From the well logging data for four drilling sites in the Janggi Basin, rudaceous sandstone and conglomerate layers were considered available $CO_2$ storage sites and the dacitic tuff and mudstone layers overlaying them as suitable cap rock [8,9]. Continuous drill cores (4.2 cm average diameter) from a drilling site 1200 m deep were acquired from KIGAM. From property analysis of these cores, three rudaceous sandstone cores (JG1-S1, JG1-S2, and JG1-S3; from 930–950 m) and three conglomerate cores (JG1-C1, JG1-C2, and JG1-C3; from 950–980 m) with average porosity of 14–18% were found and they were used for measurement of the $scCO_2$ storage ratio. For the sealing cap rock, three mudstone cores (JG1-M1, JG1-M2, and JG1-M3; from 700–760 m) and three tuff cores (JG1-T1, JG1-T2, and JG1-T3; from 800–810 m) were used for the measurement of the initial $scCO_2$ capillary entry pressure. Each rock core used in the experiment was cylindrical without cracks or fractures (4.2 cm diameter; length 5–7 cm). The geological map showing the area around the drilling site in the Janggi Basin and the rock cores used for the experiments are shown in Figure 1.

The $CO_2$ storage and sealing capacity of rocks depend on physico-chemical properties such as porosity, permeability, reaction rate, and mineralogical stability. The porosity of the rock cores was measured using the vacuum saturation method suggested by the International Society for Rock Mechanics (ISRM) with vacuum pressure of 1 torr and vacuum time of 80 min. For each sandstone and conglomerate core, several thin slabs (1 cm × 1 cm × 0.2 cm each) were also prepared to identify the mineral composition of each core by modal analysis. To quantify the average mineral portion of

each reservoir rock, 500 locations on each thin section surface of each rock slab were observed using a point-counter installed in a polarizing microscope. For each mudstone and tuff core, mineralogical and geochemical analyses were performed using XRD (X-Ray Diffractometer; X'Pert-MPD, Philips, Almelo, The Netherlands) and XRF (X-Ray Fluorescence Spectrometer; XRF-1800, Shimadzu, Kyoto, Japan), to determine their mineralogical properties.

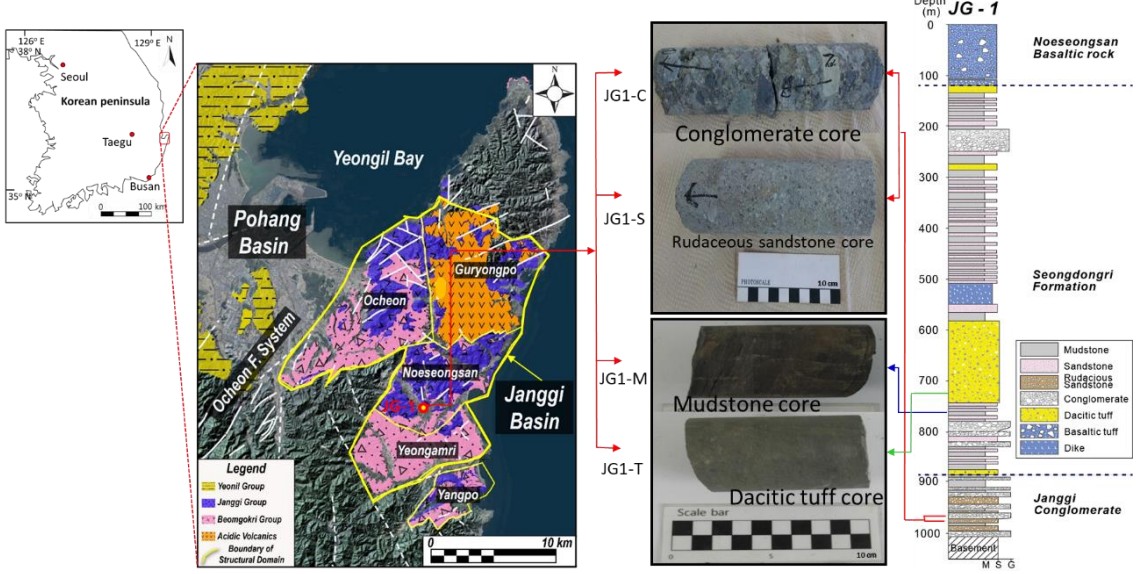

**Figure 1.** Geological map of the area around the drilling site (JG-1: **O**) in the Janggi Basin, Korea, photographs of rock cores used for the experiment and the stratigraphic columnar section (from left to right) used for the experiment (modified from [7,8]).

## 2.2. Measurement of the scCO$_2$ Storage Ratio for the Conglomerate and Sandstone Cores

When scCO$_2$ was injected, water filling the void spaces of the reservoir rock was not fully replaced by the scCO$_2$ because of the difference in interfacial tension between water and the scCO$_2$ (or difference in wettability) [24]. Thus, the amount of scCO$_2$ that could be stored in the subsurface reservoir rock was less than the total void space of the rock, and was affected by various physico-chemical parameters. For selection of an optimal CO$_2$ storage site, the scCO$_2$ storage capacity of rock from that specific reservoir should be estimated, based on the real scCO$_2$ storage ratio. Moreover, the potential amount of storage for each kind of reservoir rock should also be determined before subsurface injection of the scCO$_2$. Laboratory scale measurement of the scCO$_2$ storage ratio, displacing water from the pore spaces of Janggi sandstone and conglomerate cores under the simulated scCO$_2$ injection conditions was performed. The experimental conditions were maintained (50 °C and 100 bar) to simulate the CO$_2$ storage conditions underground. The sandstone and conglomerate cores were cut (4.2 cm in diameter and 5–7 cm in length) and their cut surfaces were polished using powdered diamond paper to maintain a uniform scCO$_2$ or water injection pressure on the cut surface. A high-pressure stainless-steel cell was developed to measure the amount of scCO$_2$ stored in the pore spaces of each core after scCO$_2$ injection. It is difficult to measure the scCO$_2$ remaining in the pore space of the rock core after scCO$_2$ injection because of the leakage of the injected scCO$_2$ at the cylinder surface boundary between the cell inner wall and the rock-core wall surface. The high-pressure cell was designed with two different walls; the inner wall was composed of a thick rubber layer (1 cm thick) and the outer wall of stainless steel. The space between the inner and the outer wall of the cell was sealed with pressurized water, which was injected from outside the cell. The surface of the inner rubber wall was in tight contact with the rock core cylinder surface, when the water pressure in the space was much higher than the scCO$_2$ injection pressure (Δp > 100 bar). The rock core top and bottom head surfaces were held using a screw-type steel holder with a hole in the middle for scCO$_2$ or water injection/drainage in the rock

core. It was possible to shut off the bypass of injected $scCO_2$ or pore water through the boundary between the core cylinder surface and the cell inner wall, allowing the $scCO_2$ (or water) to flow only through pore spaces within the rock core. Figure 2 shows photographs of the high-pressure cell and the schematic diagram for the cross-section of the cell used in the experiment.

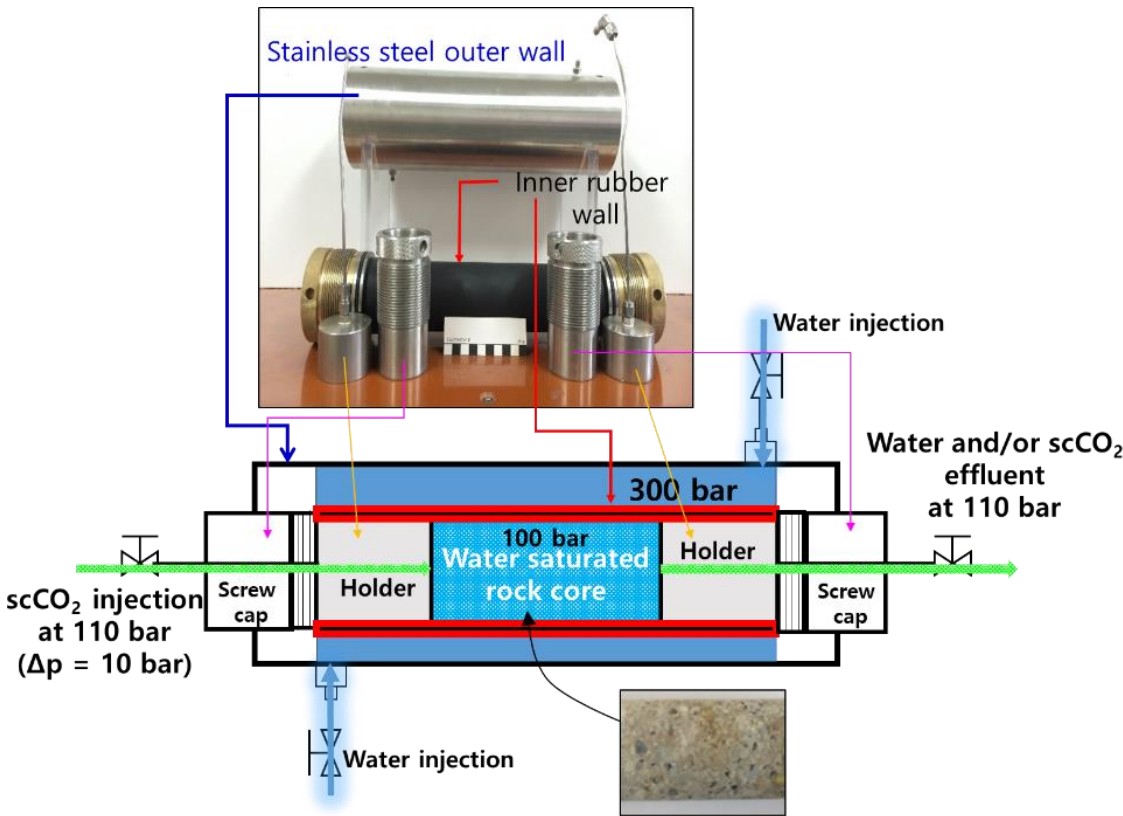

**Figure 2.** Photograph of the high-pressure cell used for the experiment to measure the supercritical $CO_2$ ($scCO_2$) storage ratio.

For the experiment, each core was fully dried at 50 °C in an oven and then weighed. The dried core was fixed by two core holders inside the high-pressure stainless-steel cell. The outer wall of the high-pressure cell was covered by a heating jacket to maintain the cell wall at the cell temperature of 50 °C. Distilled water was injected into the sealed space between the inner wall and the outer wall of the cell by a syringe pump (Isco-D260; Teledyne Isco, Inc., Nebraska, NE, USA), which was maintained at 300–350 bar. Then, the distilled water was flushed through the core at 100 bar (the injection pressure) for three pore volumes of the core to fully saturate the core with water. Next, $scCO_2$ was injected through the influent opening into the cell to displace water from the pore spaces of the core at 110 bar ($\Delta p = 10$ bar between the injection pressure and the pore water pressure in the core), while more than two pore volumes of $scCO_2$ were flushed from the core at 110 bar (assuming that displacement of water by the $scCO_2$ was successful within a few days). All of the effluent water was stored in a small stainless storage cell and its mass was weighed to measure the amount of water displaced by the $scCO_2$ in the rock core. A high-pressure stainless-steel chamber (5 L capacity) was connected to the effluent of the cell to consider the boundary condition of the reservoir rock when the $scCO_2$ was flushed from the rock core in the experiment. The water in the pores was compressed as the pore pressure increased due to $scCO_2$ injection and enough water or $scCO_2$ volume had to be provided in the chamber for the replacement of all the $scCO_2$ during the experiment. All of the high-pressure cells were maintained at 50 °C and 110 bar, after the $CO_2$ injection, to simulate the subsurface $CO_2$ storage conditions. Figure 3 shows the procedure of the experiment for $scCO_2$ exchange in the rock core.

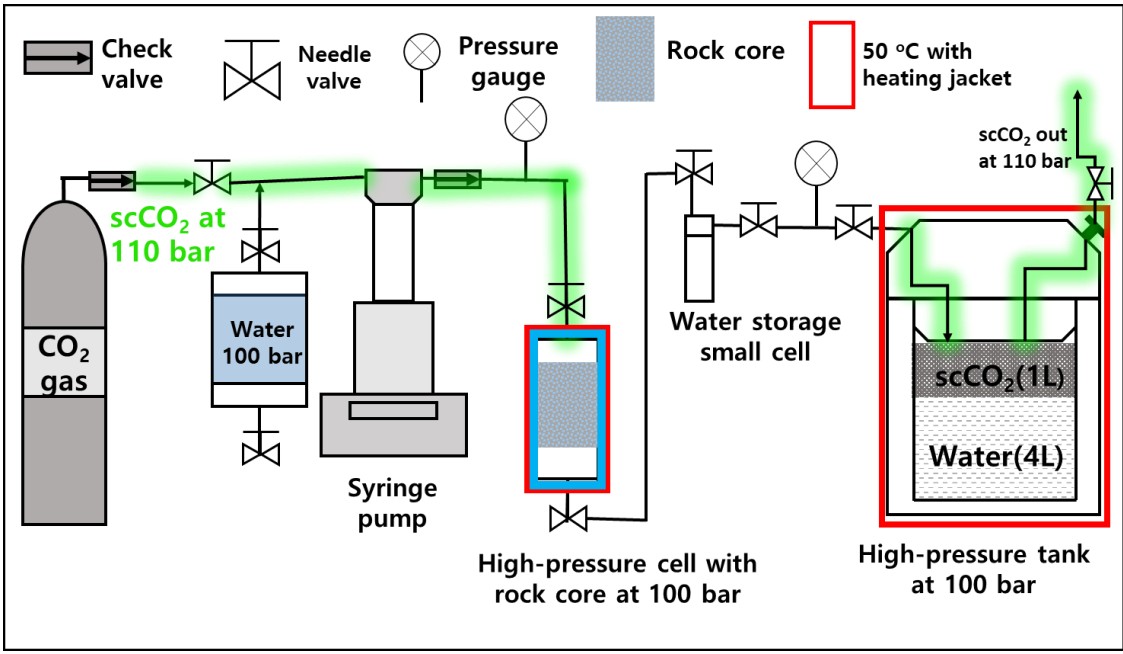

**Figure 3.** Schematic of the experiment to measure the scCO$_2$ storage ratio.

When the amount of water drained from the core was measured, the scCO$_2$ storage ratio for the specific rock core under the scCO$_2$ injection condition (in this case, 100 bar and 50 °C) could be calculated using Equation (1).

$$\text{The } scCO_2 \text{ storage ratio (\%) for the rock } (\varepsilon) = \left(1 - \frac{W_s - W_{out}}{W_s}\right) \times 100 \tag{1}$$

where $W_s$ is the volume of water saturating the core and $W_{out}$ is the water volume displaced by scCO$_2$ during the scCO$_2$ injection.

From Equation (1), the scCO$_2$ storage capacity of the conglomerate and the sandstone formations in the Janggi Basin were estimated via Equation (2) with the volume of the stratum, the average porosity, the specific gravity of the scCO$_2$, and the storage ratio of scCO$_2$ [3,14].

$$\text{The storage capacity (ton)} = \sum (V \times \varphi \times \rho \times \varepsilon), \tag{2}$$

where $V$ is the volume of the conglomerate or sandstone layer (estimated from the previous geological survey data), $\varphi$ is the average porosity, $\rho$ is the specific gravity of the scCO$_2$, and $\varepsilon$ is the scCO$_2$ storage ratio. The feasibility of the Janggi Basin as a CO$_2$ reservoir formation was evaluated based on the scCO$_2$ storage capacity of rudaceous sandstone and conglomerate drill core samples from the Janggi Basin (total of six cores).

### 2.3. Measurement of the Initial scCO$_2$ Capillary Entry Pressure for Mudstone and Dacitic Tuff

When scCO$_2$ is injected into reservoir rock, it is distributed as a separate phase from water in pore spaces and begins to move slowly upward from the lower part of the reservoir rock, due to buoyant force, because of its lower density as compared to water. Some of the injected scCO$_2$ might reach the boundary between the cap rock and the reservoir rock during continuous injection. At the early stage of scCO$_2$ injection, the amount of scCO$_2$ reaching the boundary is not much and the cap rock can prevent the intrusion of scCO$_2$ because of its low permeability. As the amount of scCO$_2$ at the boundary and its buoyant pressure increase because of continuous scCO$_2$ injection, advective and the diffusive intrusion of scCO$_2$ into the cap rock occurs. This pressure initiates seepage of the scCO$_2$ from

the reservoir rock, threatening the leakage safety of the $CO_2$ storage site. The sealing capacity of the cap rock is directly dependent on the initial $scCO_2$ seepage (or intrusion) pressure on the cap rock surface [22]. The direct measurement of the initial $scCO_2$ capillary entry pressure of the cap rock was performed at a laboratory scale, to evaluate the $scCO_2$ shielding capacity of the cap rocks in the study area. The mudstone and dacitic tuff rock cores sampled during the deep drilling expedition (from 700–800 m) at the onshore site in the Janggi Basin were used for the experiment (Figure 1).

Rock cores without cracks or fractures were cut (4.2 cm diameter, 5–6 cm length) and were then fully dried at 50 °C in an oven for 7 d. Each rock core was fixed in the high-pressure stainless-steel cell (which was used in the same way as in the previous experiment, Section 2.2). Each core was saturated with distilled water at 100 bar of pore water pressure. The effluent of the cell was connected to a large tank filled with 3 L of water and 2 L of $scCO_2$ at 100 bar and 50 °C, simulating the subsurface $scCO_2$ injection conditions. The initial $scCO_2$ capillary entry pressure into the rock core head (higher than 100 bar: $\Delta p$ = injection pressure − 100 bar), was controlled at the influent port with the regulator of the core holder in the cell bottom, until the $scCO_2$ began to penetrate the rock. The $scCO_2$ injection pressure was set at 110 bar and the injection pressure was increased by 10 bar until the $scCO_2$ began to penetrate the rock core head. At the outset, the $scCO_2$ injection pressure on the core head surface was set at 110 bar ($\Delta p$ = 10 bar) and any $scCO_2$ intrusion into the core was observed for 10 days. If no $scCO_2$ intrusion occurred, the injection pressure was increased by 10 bar for 10 more days, to monitor any $scCO_2$ intrusion into the core. This process was repeated until the $scCO_2$ began to penetrate the rock core head. When the $scCO_2$ begin to intrude and the $scCO_2$ injection pressure started to decrease, the $scCO_2$ injection pressure was maintained until $scCO_2$ was flushed from the end of the rock core. This pressure was regarded as the initial $scCO_2$ capillary entry pressure ($\Delta p$) of the rock core. The $scCO_2$ shield capacity of each kind of cap rock core (dacitic tuffs and mudstones here) was evaluated by comparing their initial $scCO_2$ capillary entry pressures ($\Delta p$).

The mineralogical changes of mudstone and tuff were also measured to evaluate their geochemical stability for 90 d of the $scCO_2$–water–rock reaction under $CO_2$ storage conditions (100 bar and 50 °C). The rock core was pulverized using a mortar, and ten grams of powdered rock materials were mixed with 100 mL of distilled water in a high-pressure stainless-steel cell (capacity of 150 mL), in which the inner wall was coated with a Teflon layer. The void spaces in the cell were filled with $scCO_2$ using a syringe pump. Then $scCO_2$–water–rock reactions were allowed to occur in the cell at 100 bar and 50 °C, simulating the subsurface storage conditions. The total reaction time was 90 d and XRD analysis was conducted before and after the reaction to identify any mineralogical changes of the cap rock due to the $scCO_2$–water–rock reaction.

## 3. Results and Discussion

### 3.1. Measurement of the $scCO_2$ Storage Ratio for the Conglomerate and Sandstone Cores

Results of the modal analyses for the rudaceous sandstone and conglomerate cores are shown in Table 1 and the photomicrographs of their thin sections are shown in Figure 4. The conglomerate was mostly composed of rhyrolitic and andesitic rock fragments (average 74.5%), followed by quartz, clay/accessory minerals, feldspar, micas, and calcite (in descending order). The sandstone mainly consisted of quartz, rock fragments, clay/accessory minerals, feldspar, micas, and calcite. Their average proportions were 32.5%, 23.8%, 18.3%, 19.4%, 2.5%, and 2.5%, respectively (Table 1). In previous studies it was observed that calcite, feldspars, chlorite, micas, and clay minerals bearing Ca and Mg might control the geochemical reactions with $CO_2$ in the storage site, thereby regulating the physical properties of the reservoir rock [25–28]. The results from the XRD analyses (not shown in this paper) showed a mineral composition similar to that indicated by the modal analysis.

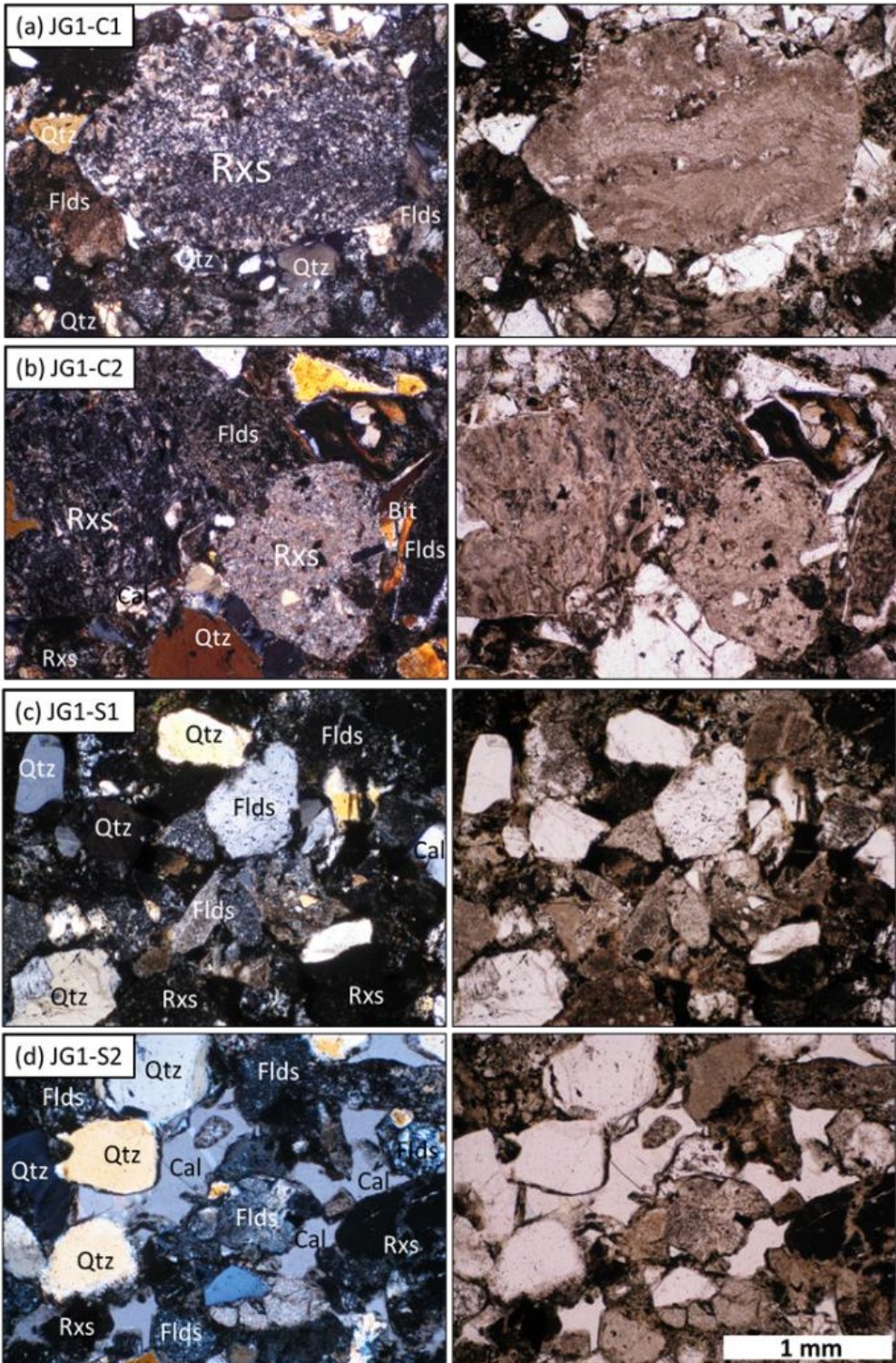

**Figure 4.** Photomicrographs of thin sections of the conglomerate (**a,b**) and the rudaceous sandstone (**c,d**); left column—closed mode; right column—open mode; Qtz—quartz; Flds—feldspars; Cal—calcite; Bit—biotite; and Rxs—rock fragments.

The porosity of the conglomerate and the sandstone cores were measured and are shown in Table 2. The average porosity of the conglomerate and the rudaceous sandstone was 17.8% and 14.5%, respectively, suggesting that they fall within the porosity range of typical $CO_2$ storage formations in other places, where large amounts of scCO$_2$ have been injected [22,29–34]. The scCO$_2$ storage ratio

of each conglomerate and sandstone core was measured and their averages are shown in Table 2. The calculated average scCO$_2$ storage ratio of the Janggi conglomerates was 30.7%, suggesting that 30.7% of the void space in the conglomerate was filled with scCO$_2$, while the pressure difference between the scCO$_2$ injection and the pore water was maintained at 10 bar. The average scCO$_2$ storage ratio of rudaceous sandstones was 13.0%, which was about two-fifths that of the conglomerate. From these results, both kinds of rock had great capability for storing CO$_2$ in their pore spaces, but the conglomerate was considered to be a better option for a CO$_2$ storage site than the rudaceous sandstone.

**Table 1.** Petrographic detrital modal analysis of the rock cores.

| Rock Type | Mineral Portion (%) | | | | | | | |
|---|---|---|---|---|---|---|---|---|
| | Quartz | K-Feldspar | Plagioclase | Rock Fragment | Fine Matrix and Clay | Calcite | Micas | Others |
| JG1-C1 | 7.8 | 1.8 | 1.2 | 80.6 | 8.0 | 0.2 | 0.0 | 0.4 |
| JG1-C2 | 17.8 | 6.6 | 4.8 | 57.4 | 10.8 | 0.4 | 1.2 | 1.0 |
| JG1-C3 | 6.4 | 1.8 | 1.0 | 85.4 | 3.8 | 0.4 | 0.4 | 0.8 |
| Average ± standard deviation | 10.7 ± 6.2 | 3.4 ± 2.8 | 2.3 ± 2.1 | 74.5 ± 15.0 | 7.5 ± 3.5 | 0.3 ± 0.1 | 0.5 ± 0.6 | 0.7 ± 0.3 |
| JG1-S1 | 33.6 | 8.8 | 12.0 | 24.8 | 16.0 | 2.2 | 1.2 | 1.4 |
| JG1-S2 | 31.6 | 11.6 | 8.2 | 24.2 | 18.4 | 0.6 | 4.4 | 1.0 |
| JG1-S3 | 32.2 | 10.4 | 7.0 | 22.4 | 20.4 | 4.6 | 1.8 | 1.2 |
| Average ± standard deviation | 32.5 ± 1.0 | 10.3 ± 1.4 | 9.1 ± 2.6 | 23.8 ± 1.2 | 18.3 ± 2.2 | 2.5 ± 2.0 | 2.5 ± 1.7 | 1.2 ± 0.2 |
| JG1-M1 | 40.8 | 3.0 | 1.4 | 0.4 | 47.4 | 1.2 | 2.6 | 3.2 |
| JG1-M2 | 43.8 | 2.8 | 0.6 | 0.0 | 47.4 | 0.4 | 2.4 | 2.6 |
| JG1-M3 | 33.6 | 2.4 | 0.4 | 0.2 | 58.6 | 0.2 | 2.2 | 2.4 |
| Average ± standard deviation | 39.4 ± 5.2 | 2.7 ± 0.3 | 0.8 ± 0.5 | 0.2 ± 0.2 | 51.1 ± 6.5 | 0.6 ± 0.5 | 2.4 ± 0.2 | 2.7 ± 0.4 |
| JG1-T1 | 26.0 | 12.2 | 11.6 | 18.6 | 19.6 | 1.4 | 3.8 | 6.8 |
| JG1-T2 | 19.4 | 10.4 | 14.8 | 11.6 | 32.4 | 1.2 | 4.2 | 6.0 |
| JG1-T3 | 24.5 | 9.7 | 13.2 | 16.7 | 22.0 | 2.4 | 3.1 | 8.4 |
| Average ± standard deviation | 23.3 ± 3.5 | 10.8 ± 1.3 | 13.2 ± 1.6 | 15.6 ± 3.6 | 24.7 ± 6.8 | 1.7 ± 0.6 | 3.7 ± 0.6 | 7.1 ± 1.2 |

* JG1-C—conglomerate cores; JG1-S—rudaceous sandstone cores, JG1-M—mudstone cores and JG1-T—dacitic tuff cores.

**Table 2.** The scCO$_2$ storage ratios of conglomerate and rudaceous sandstone cores at the Janggi Basin.

| Core Number | Rock Type | Porosity (%) | The scCO$_2$ Storage Ratio (%) |
|---|---|---|---|
| JG1-C1 | | 19.23 | 23.56 |
| JG1-C2 | Conglomerate | 18.92 | 35.41 |
| JG1-C3 | | 15.12 | 31.20 |
| Average | | 17.76 | 30.72 |
| JG1-S1 | | 10.54 | 13.12 |
| JG1-S2 | Rudaceous sandstone | 16.44 | 16.28 |
| JG1-S3 | | 17.26 | 9.64 |
| Average | | 14.75 | 13.01 |

The scCO$_2$ storage ratio values were used for the estimation of the CO$_2$ storage capacity for the Janggi Basin (Table 3). From previous studies [7–9] it was known that the Janggi Basin extends horizontally over about 0.25 km$^2$ below a depth of 800 m and that the thickness of both the conglomerate and sandstone layers available for the CO$_2$ reservoir is 100 m (about 1:1 ratio), assuming that the maximum volume of the CO$_2$ storage formation in the Janggi Basin under 800 m in depth is about

0.025 $km^3$. The parameter values used to calculate the $CO_2$ storage capacity and the calculated $scCO_2$ capacity for the two formations are shown in Table 3. The $scCO_2$ storage capacity of the reservoir rocks around the probable $scCO_2$ injection site in Janggi Basin was calculated to be 368,742 metric tons, demonstrating that the conglomerate and sandstone formations in Janggi Basin have great potential for use as a pilot test site for $CO_2$ storage in Korea (to receive more than 100,000 metric tons of $CO_2$ injection at the test storage site). As mentioned earlier, the amount of dissolved $CO_2$ in the pore water of each core was ignored while calculating the amount of $scCO_2$ storage in Table 3. When the amount of dissolved $CO_2$ in the pore water was considered, the already substantial $CO_2$ storage capacity of the Janggi Basin was increased. Thus, if enough $scCO_2$ was secured in the conglomerate and the sandstones, the Janggi Basin could be evaluated as a positive storage site, regardless of the amount of $CO_2$ dissolved in the pore water.

**Table 3.** The $scCO_2$ storage capacity of conglomerate and rudaceous formations at the Janggi Basin (unit—metric ton).

| Rock Type | The Calculated Amount of $scCO_2$ Storage for A Conglomerate and Rudaceous Formation (metric ton) |
|---|---|
| Conglomerate | V(50 m × 250 m × 1000 m) × φ(0.1776) × ρ*(400 $kg/m^3$) × ε(0.3072) = 272,793.6 |
| Rudaceous sandstone | V(50 m × 250 m × 1000 m) × φ(0.1475) × ρ*(400 $kg/m^3$) × ε(0.1301) = 95,948.8 |

\* Density of $scCO_2$ at 100 bar and 50 °C (from [35,36]).

### 3.2. Measurement of the Initial $scCO_2$ Capillary Entry Pressure for Mudstone and Dacitic Tuff

The mineralogical changes of the mudstone and the dacitic tuff after 90 days of $scCO_2$–water–rock reaction were investigated by XRF analyses and their results are shown in Table 4. The results indicated that even after 90 d of reaction, the proportions of the major constituents of the two kinds of rock were not significantly changed, except for a minor decrease of $SiO_2$ and CaO. These originated from increased dissolution of calcite, Ca–feldspar, and Ca-bearing silicates, which were similar to results of previous studies [37–39]. It is suggested that the mudstone and the dacitic tuff in the Janggi Basin is likely to maintain significant stability against $scCO_2$-involved geochemical reactions during $CO_2$ storage.

**Table 4.** XRF analysis of mudstone and dacitic tuff before and after 90 days of $scCO_2$–water–rock reaction.

| Composition | Ratio (wt. %) | | | | | |
|---|---|---|---|---|---|---|
| | JG1-M1 | | JG1-M2 | | JG1-T1 | |
| | Before | After 90-day Reaction | Before | After 90-day Reaction | Before | After 90-day Reaction |
| $SiO_2$ | **57.20** | **56.51** | **56.36** | **55.65** | **57.01** | **56.61** |
| $Al_2O_3$ | 20.84 | 20.89 | 17.88 | 17.94 | 18.21 | 18.33 |
| TiO2 | 0.78 | 0.78 | 0.62 | 0.62 | 0.76 | 0.76 |
| $Fe_2O_3$ | 5.32 | 5.29 | 4.99 | 5.06 | 7.04 | 7.10 |
| MnO | 0.09 | 0.08 | 0.12 | 0.12 | 0.15 | 0.15 |
| MgO | 0.79 | 0.92 | 0.71 | 0.81 | 1.95 | 2.04 |
| CaO | **1.14** | **1.08** | **1.16** | **1.09** | **5.03** | **4.64** |
| $Na_2O$ | 1.63 | 1.73 | 1.32 | 1.36 | 2.55 | 2.60 |
| $K_2O$ | 2.30 | 2.40 | 2.04 | 2.09 | 0.95 | 0.94 |
| $P_2O_5$ | 0.09 | 0.09 | 0.15 | 0.15 | 0.05 | 0.05 |
| LOI | 9.61 | 10.07 | 14.44 | 14.96 | 6.12 | 6.58 |
| Total | 99.80 | 99.85 | 99.79 | 99.85 | 99.82 | 99.80 |

Results from the measurement of the initial $scCO_2$ capillary entry pressure (Δp) for the mudstone and the tuff core are shown in Figure 5. For all of tuff cores, the $scCO_2$ began to intrude into the rock core at 115 bar (Δp = 15 bar) and continuous $scCO_2$ injection into the core occurred at a Δp higher

than 20 bar. This suggests that the initial scCO$_2$ capillary entry pressure ($\Delta$p) of the dacitic tuff ranged from 15 to 20 bar, under the conditions of 100 bar and 50 °C. In the tuff cores, 8–10% of the void space (0.7–0.9 mL) was filled by scCO$_2$ at a $\Delta$p higher than 20 bar. In a previous study (Kim et al. 2019), the average initial scCO$_2$ capillary entry pressure of the sandstone and the conglomerate in the Janggi Basin was lower than 10 bar (mostly < 5 bar), which was less than one-third that for the dacitic tuff. For the mudstone cores in the Janggi Basin, the scCO$_2$ did not penetrate the core surface and the stored scCO$_2$ was less than 0.005 mL, even when the injection pressure was 250 bar ($\Delta$p = 150 bar) for 30 d. This suggests that the initial scCO$_2$ capillary entry pressure for the mudstone core was much higher than 150 bar (10 times higher than for the tuff). Based on the initial scCO$_2$ capillary entry pressure, the mudstone formation in the Janggi Basin was much more suitable than the tuff formation as a shield against scCO$_2$ leakage from the reservoir rock.

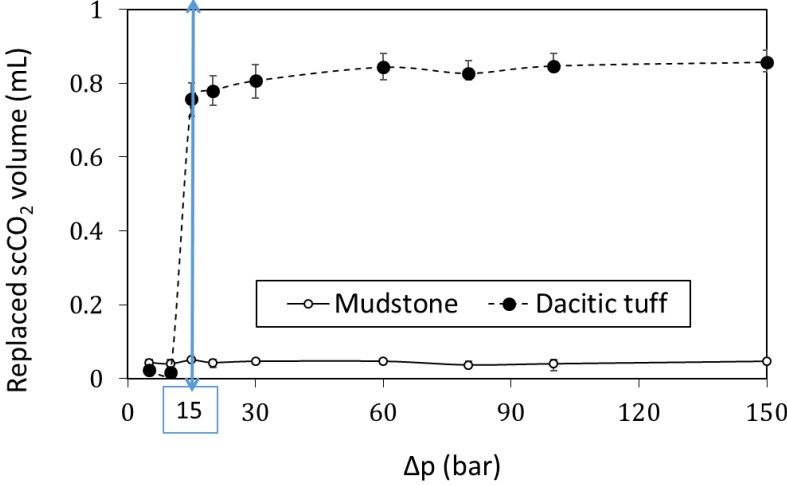

**Figure 5.** Initial scCO$_2$ capillary entry pressure ($\Delta$p) and the volume of scCO$_2$ stored in mudstone and dacitic tuff cores.

## 4. Conclusions

Determining an optimal storage site and choosing the main parameters to be considered has been one of the main issues for the geological sequestration of CO$_2$. During the last two decades, evaluation of the storage capacity and the leakage safety has been considered essential for optimal storage site selection. However, previously, only rough estimations were made of the reservoir and the capping rock using conventional parameters such as porosity and permeability, or by using large-scale information acquired from geophysical exploration and geological field observation. The quantitative evaluation of CO$_2$ storage capacity and of the risk of CO$_2$ leakage has been very limited, even at a laboratory scale. Recently, a direct technique for the measurement of the scCO$_2$ storage ratio and the initial scCO$_2$ capillary pressure of the rock was developed, but its application to CO$_2$ storage site selection has almost never been attempted in previous studies. As mentioned earlier, the Janggi Basin was selected as a feasibility testing site to store at least 100,000 metric tons of CO$_2$ before starting large-scale injection (more than 1,000,000 metric tons) on the Korean peninsula. For the testing site, it was important that the CO$_2$ storage capacity of the Janggi Basin was, at the least, larger than 100,000 metric tons. This study presents an easy and effective technique by which to evaluate the CO$_2$ capacity of reservoir rock and the leakage safety of the cap rock, and finally, to successfully select the Janggi basin as an optimal CO$_2$ storage site in Korea. The scale-up estimation of the CO$_2$ storage capacity for the conglomerate and the rudaceous sandstone in the Janggi Basin of Korea was performed on the basis of direct measurement of the scCO$_2$ storage ratio. The safety risk of scCO$_2$ leakage for the cap rock of dacitic tuff and mudstone in the Janggi Basin was also quantitatively evaluated by measuring the initial scCO$_2$ capillary entry pressure.

The experimental results successfully demonstrated that the conglomerate and sandstone formations of the Janggi Basin are suitable as a geological storage test site, for injection of a hundred thousand tons of $CO_2$ from the viewpoint of storage capacity. If the amount of dissolved $CO_2$ in the pore water could be considered, a more precise estimate of the $CO_2$ storage capacity for the specific reservoir formation could be estimated; related research in this area is already in progress.

It was also verified that the mudstone formation in the Janggi Basin is adequate to prevent the seepage of buoyant $scCO_2$ from the reservoir site because its initial capillary entry pressure ($\Delta p$) was higher than 150 bar. From the XRF analysis before and after the experiment, reliable evidence for the geochemical stability of the tuff and the mudstone was also provided. These quantitative measurements of the $scCO_2$ storage ratio and the initial $scCO_2$ capillary entry pressure applied in this study could be used to determine practical $CO_2$ storage sites and could also provide meaningful information for future decisions regarding $scCO_2$ injection conditions.

This study focused on the hydromechanical measurement of the $scCO_2$ in the pore spaces of rock core, over just a few months and, thus, did not deal with the effect of geochemical reactions ($CO_2$–water–rock) on the $scCO_2$ capacity and the capillary entry pressure with a long-term view. Recent research indicates that physical or chemical changes in the properties of rock involved in $CO_2$ sequestration could arise from mineralogical or geochemical reactions within a shorter time after $CO_2$ injection, than previously expected. For successful selections of optimal $CO_2$ storage subsurface sites, the geochemical stability of reservoir or cap rocks should be considered an important parameter, along with the storage capacity and the capillary entry pressure.

**Author Contributions:** M.L. and J.P. conceived and designed the methodology and the experiments; J.P., M.L., and S.K. performed the experiments and the data analyses; S.W. and M.Y. contributed materials and data interpretation; M.L. wrote the paper to prepare the submission.

**Funding:** This research was supported by the Korea Agency for Infrastructure Technology Advancement (KAIA) grant funded by the Ministry of Land, Infrastructure and Transport (Grant 19CTAP-C151965-01) and by the grant (2019002470002) from Korea Ministry of Environment as "The SEM (Subsurface Environmental Management) projects".

**Acknowledgments:** The authors would like to express their gratitude to the anonymous reviewers for their critical comments and advice.

**Conflicts of Interest:** The authors declare no conflict of interest.

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
