# Peer review of "Estimates of scCO2 Storage and Sealing Capacity of the Janggi Basin in Korea Based on Laboratory Scale Experiments"

_minerals, doi:10.3390/min9090515_

Round 1

Reviewer 1 Report

Thank you for giving me the opportunity to review this very good article for which I have only one very minor comment;

p9 line 291, is there any reference of previous work on studying geochemical reactions with similar type of rock existing? Just to confirm your point (for which I agree too)

Best wishes

Author Response

For Reviewer 1:

Thank you for giving me the opportunity to review this very good article for which I have only one very minor comment;

Question (1): p9 line 291, is there any reference of previous work on studying geochemical reactions with similar type of rock existing? Just to confirm your point (for which I agree too)

Answer (1): Thanks for good suggestion. Research papers for the scCO2 reactions with cap rocks are relatively rare than those with storage rocks such as sandstones. We cannot find out previous papers studying on the scCO2 related geochemical reaction with the identical tuff and mudstone at the Janggi Basin, Korea. But, recently, there are several papers that deal with the scCO2-groundwater-clay (or clay minerals) reactions for the cap rocks (listed below). At the reviewer’s suggestion, the latest references related to the geochemical reactions with similar clays of cap rocks were added in the manuscript and in reference list. (Line 377 – 378; reference [37, 38, 39])

- Eszter Sendula et. al. (2017) Experimental study of CO2-saturated water-illite/kaolinite/montmorillonite system at 70-80 oC, 100-105 bar, Energy Procedia (114), p4934-4947. (http://creativecommons.org/licenses/by-nc-nd/4.0/)

- Reza Rezaee et. al. (2017) Shale alteration after exposure to supercritical CO2, International Journal of Greenhouse Gas Control (62), p91-99. (http://dx.doi.org/10.1016/j.ijggc.2017.04.004)

- Xiangrong Luo et. al. (2018) Supercritical CO2-water-shale interactions under supercritical CO2 simulation Conditions, Energy Procedia (144), p182-185. (10.1016/j.egypro.2018.06.024)

Reviewer 2 Report

The comments in the word file attached.

Author Response

For Reviewer 2:

 Question: At the first glance this paper seems to be suitable for the MDPI Minerals Journal. Its topic - the CO2 storage is still sound, and the work is based on well interpreted laboratory experiments. Unfortunately the presented results were compiled from 3 previous papers (it must be admitted however, that they were cited in the manuscript):

An, J.; Lee, M.; Wang, S. Evaluation of the Sealing Capacity of the Supercritical CO2 by the Measurement of Its Injection Pressure into the Tuff and the Mudstone in the Janggi Basin. Econ. Environ. Geol. 2017, 50, 403 303–311. Wang, S.; Kim, J.; Lee, M. Measurement of the scCO2 Storage Ratio for the CO2 Reservoir Rocks in Korea. Proceedings of the Energy Procedia 2016, 97, 342–347, doi:10.1016/j.egypro.2016.10.015. Kim, S.; Kim, J.; Lee, M.; Wang, S. Evaluation of the CO2 Storage Capacity by the Measurement of the scCO2 . Displacement Efficiency for the Sandstone and the Conglomerate in Janggi Basin. Econ. Environ. Geol. 2016, 366 49, 469–477.

Question (1): Paper 1 (Abstract) - one of the main conclusions: “The initial scCO2 injection pressure (ΔP) of a tuff in the Janggi basin was 15 bar and the continuous scCO2 injection into the tuff core occurred at higher than 20 bar of injection pressure. For the mudstone in the Janggi basin, the initial scCO2 injection pressure was higher than 150 bar (10 times higher than that of the tuff). From the results, the mudstone in Janggi basin was more suitable than the tuff to shield the scCO2 leakage from the reservoir rock at subsurface.” The reviewed manuscript (second of the two main conclusions) – the text identical as above.

Answer (1): Thanks for the reviewers’ suggestion. The paragraph in abstract was fully modified.

From “The initial scCO2 injection pressure (ΔP) of a tuff in the Janggi basin was 15 bar and the continuous scCO2 injection into the tuff core occurred at higher than 20 bar of injection pressure. For the mudstone in the Janggi basin, the initial scCO2 injection pressure was higher than 150 bar (10 times higher than that of the tuff). From the results, the mudstone in Janggi basin was more suitable than the tuff to shield the scCO2 leakage from the reservoir rock at subsurface.”

To “The scCO2 capillary entry pressure for the tuff ranged from 15 to 20 bar and for the mudstone it was higher than 150 bar, suggesting that the mudstone layers have enough sealing capacity from the aspect of hydromechanics. From XRF analyses before and after 90 days of the scCO2-water-cap rock reaction, the mudstone and the tuff were investigated to maintain the geochemical stability as the cap rock. From the study, the Janggi Basin is considered an optimal CO2 storage site based both on its high scCO2 storage ratio and high capillary entry pressure”. (Please, see the Abstract)

Question (2): Paper 2 and Paper 3– storage capacities were calculated based on the same methodology, assumptions and very similar core data as in the reviewed manuscript, and delivered similar results, see the comparison:

Paper 2 (Table 2 and 3):

Paper 3 (Table 2 and 3):

The reviewed manuscript (Table 2 and 3):

Answer (2): For the paper, there are very different viewpoints between authors and Reviewer. At first, ① there are no duplication of identical data in papers in Table 2 and 3. All of rock cores used in this paper were different to those used in previous papers. Even formats in Table of the submitted paper look similar to those in previous papers, identical data and values for porosity duplicated in papers do not exist (Please, compare values each other in Table 2 and 3). ② In previous papers, the volume of storage formation in the Janggi Basin was simply assumed as the 50 m x 250 m x 250 m (unfounded assumption by the author). However, in submitted paper, the amount of scCO2 in the Janggi basin was calculated based on the 3 dimensional volume which was referred from the recent geological survey and field study (Line 44-54; Line 299-304; ref. [7,8,9]). Thus, by application of different cores and more realistic subsurface volumes of the Janggi Basin, the more realistic estimation for the scCO2 storage capacity of the Janggi Basin could be complemented in this study (Table 3). ③ Also, in this study, several experimental conditions for the scCO2 storage ratio and the capillary entry pressure measurement such as the boundary condition of the high pressurized tank and the scCO2 injection time on the core surface were also modified to more realistically simulate the scCO2 leakage at the boundary between the reservoir and the cap rock (Line 103-107; Figure 3). Authors are confident that all these reasons clearly differentiate data in the submitted paper from those in previous papers. Authors revised the original paper to emphasize these differences and added many new sentences in “Introduction” and “Conclusions” sections. Authors earnestly ask for reviewers’ understanding.    

Question (3) Conclusion: The reviewed paper has been compiled from 3 previous published papers, and does not provide enough new information, and as such should be not accepted for publication in the MDPI Minerals Journal. I encourage the authors to develop a broader interpretation of their research conducted in Janggi Basin, and to present new, previously unpublished ideas and data.

Answer (3): Even there are different viewpoints of this paper between authors and Reviewer 2, authors tried to respond to all of questions in detail.

As the reviewers’ suggestion, authors sharply revised the manuscript to emphasize the differences between this paper and previous papers and included several paragraphs highlighting the novelty and originality of the paper. Detail description about results in three previous papers and about main differences between the paper and the previous were newly added in manuscript. Only for the comparison with the previous and for the novelty of the paper, one page volume was newly added mainly in “Introduction” and “Conclusion” section and more than 30 % of the original paper was fully revised. Please, see “Abstract”, “Introduction” and “Conclusion” sections for further review. All revised and added parts were in red color and were also reviewed by the proofreader in English.

Results obtained from the similar measurement technique developed in the previous study can have novelty and originality. The previous papers focused on the development of the hydromechanical measurement technique for the scCO2 storage ratio and the capillary entry pressure in the laboratory. Their results were still in the fundamental experiment stages and does not include any mineralogical and geochemical approach to select the practical CO2 storage site. Also, in previous papers, there does not exist any connection between the scCO2 storage ratio and the capillary entry pressure as the parameters to select optimal CO2 storage site. However, the submitted paper focused on the whole process to select the optimal CO2 storage site and this process includes not only the quantitative measurement of both for the scCO2 storage and sealing capacity but also the application of the geological survey data and the mineralogical and geochemical properties. This study presents a novel and reliable laboratory scale document by which to select a successful CO2 storage site based on both of quantitative scCO2 storage ratio and capillary entry pressure measurement under CO2 sequestration conditions as well as on geochemical analyses. The well-formed application like this in the laboratory scale for the successful CO2 site selection has never been attempted. By these reasons, authors believe this revised paper contains more meaningful data and enough originality than previous papers (It is not simply compiled from previous papers). If the reviewer still does not agree with authors’ opinion. Please, support parts that the reviewer disagrees with authors and authors may ask the editor argue this case with other reviewers. Authors earnestly ask for reviewers’ understanding.

Here are newly added paragraphs and sentences to emphasize the differences between this paper and previous papers and to highlight the novelty and originality of the paper.

“From previously collected well-logging data and the geo-structural interpretation results, the Janggi conglomerate formation in the Janggi Basin can be divided into 4 lithofacies. These consist of conglomerate and rudaceous sandstone from gravelly braided stream deposits, coarse sandstone deposited in mouth-bar or delta, muddy sandstone and shale deposited in floodplain environments, and mudstone as lacustrine deposits [8]. The western part of the basin is mainly composed of thick conglomerate and rudaceous sandstone lithofacies, which are available for use as a CO2 storage reservoir, whereas the mudstone and muddy sandstone lithofacies constitute the eastern part of the basin. The western part of the Janggi Basin without large faults is considered an optimal CO2 storage area and its area is now estimated to be at least about 10 km2, assuming that the practical volume of the CO2 storage volume in the Janggi Basin deeper than 800 m is about 0.025 km3 [8, 9].” (Line 44 – 54 in “Introduction”)

“In 2016, Wang et al. and Kim et al. [19, 20] carried out direct laboratory measurement of the scCO2 storage ratio for possible reservoir rock cores, presenting the possibility that it might be used to estimate the scCO2 capacity for the specific reservoir. However, this process was still in the experimental stages and the number of studies on scCO2 storage capacity based on direct measurement in the laboratory is very limited.” (Line 78 – 82 in “Introduction”)

“The estimation of practical CO2 storage amount for a specific reservoir rock could be made possible by using both the scCO2 storage ratio and the reservoir volume acquired from geological survey, which has almost never been tried before. In this study, laboratory experiments were performed to measure the amount of scCO2 displacing water from the pore spaces of the sandstones and conglomerate cores sampled from 800–1000 m depth in the Janggi Basin, which is classified as an available CO2 storage reservoir in Korea. The scCO2 storage capacity of the Janggi Basin was calculated quantitatively according to the measured scCO2 storage ratio and with additional geophysical data on the spatial domain of the Janggi Basin.” (Line 86 – 94 in “Introduction”)

“Even if a CO2 storage site has enough scCO2 storage capacity, it has to be excluded from suitable CO2 storage sites if the scCO2 leakage safety of the cap rock in the site is not assured.” (Line 97 – 98 in “Introduction”)

“In 2017, the initial capillary entry pressure of scCO2 into the cap rock core surface, determined when the scCO2 began to infiltrate the rock, was successfully measured in the laboratory [23]. In this study, several experimental conditions for the capillary entry pressure measurement such as the boundary condition of the high pressure tank and the scCO2 injection time on the core surface were modified to more realistically simulate the scCO2 leakage at the boundary between the reservoir and the cap rock. More information for the comparison of experimental conditions can be drawn from [23].” (Line 101 – 107 in “Introduction”)

“From the experimental results on the scCO2 storage ratio for the reservoir rock, on the initial scCO2 capillary entry pressure for the cap rock, and from mineralogical analyses for rock cores, the feasibility of the Janggi Basin as an available pilot-scale CO2 storage test site where a hundred thousand metric tons of CO2 could be injected was evaluated. This study presents a novel and reliable method by which to select a successful CO2 storage site based on both quantitative scCO2 storage ratio and capillary entry pressure under CO2 sequestration conditions, as well as on geochemical analyses.” (Line 112 – 118 in “Introduction”)

“Recently, a direct technique for the measurement of the scCO2 storage ratio and the initial scCO2 capillary pressure of the rock was developed but its application to the CO2 storage site selection has almost never been attempted in previous studies. As mentioned in the Introduction, the Janggi Basin was selected as a feasibility testing site to store at least 100,000 metric tons of CO2 before starting large scale injection (more than 1,000,000 metric tons) on the Korean peninsula. For the testing site, it is important that the CO2 storage capacity of the Janggi Basin is, at the least, larger than 100,000 metric tons. This study presents an easy and effective technique by which to evaluate the CO2 capacity of reservoir rock and the leakage safety of the cap rock, and finally to successfully select the Janggi basin as an optimal CO2 storage site in Korea.” (Line 357 – 365 in “Conclusions”)

“From XRF analysis before and after the experiment, reliable evidence for the geochemical stability of the tuff and the mudstone was also provided.” (Line 377 – 378 in “Conclusions”)

 ------------------------------------- The end ---------------------------------

Reviewer 3 Report

Review of Park et al., The scCO2 storage and sealing capacity of the Janggi Basin in Korea; based on laboratory experiments

Submitted for publication in minerals

In their proposed article the authors study the important question of how much CO2 could be stored safely with in the the Janggi Basin, South Korea. They use laboratory experiments to evaluate the storage capacity for CO2 storage of sandstones and conglomeratic formations as well as experiments which analyse the sealing potential of the caprocks occurring in the basin. After discussing the experimental results, the authors conclude, that the conglomerates in the basin are an ideal storage formation and that enough storage space for a test injection site.

While the manuscript covers an important part of any CO2 storage operation – identifying the amount of CO2 that can be safely stored within the reservoir – I find that several key issues need to be addressed before I can recommend publication. These are:

-       The geological setting of the basin is not very well described. A stratigraphic column is a must, and if possible a cross section. At the moment it is not clear how the subsurface is structured. Are there any faults? The basin is very small (10 km2), how would this affect the pressure distribution during injection? Does the pore-water have space to escape to? How are the edges of the basin?

-       The authors use the term seepage pressure several to describe how CO2 migrates into or through caprocks. A much better term would be capillary entry pressure and capillary breakthrough pressure – these are widely used in the CO2 storage community (e.g. Iglauer, 2018, https://doi.org/10.1016/j.ijggc.2018.07.009, or Miocic et al., 2019, https://doi.org/10.5194/se-10-951-2019) and are best used to differentiate between the different pressures.

-       The discussion is lacking several important issues: a) In the experiments destilled water was used, but other experiments have highlighted that the salinity of brine has a big impact on the capillary storage capability for CO2. b) It is unclear if CO2 dissolution into the pore water and residually trapped CO2 was considered when the storage capacity was calculated. During the experiments some volume of CO2 must have been residually trapped? In the introduction it is explained that the dissolved CO2 is neglected, as the direct storage is seen as “the substantive amount of CO2”. How much is that exactly?

-       The description of the methods is not consistent, for some experimental steps the time specifications are not mentioned.

-       The reservoir rocks consist of many rock fragments, fine matrix, clays and feldspars – while the authors have not observed much of an alteration during the 90 days of experiment, this amount of reactive minerals could be an issue during long-term (10.000 years) storage? A discussion on this is needed.

-       Most of the paper is written well, however, the abstract needs some significant rewriting with regards to the English language.

Some more minor issues:

- Line 35-36: Reference to figure 1 should be places here

- Line 44-45: Here it should be mentioned how the basin is separated from neighbouring areas.

-Line 76: m depth

-Line 80-82: For me duration of sequestration means the time of injection, but the storage site should be secure for the whole lifespan of at least 10.000 years.

-Line 84 and onwards: Replace initial seepage pressure with the correct capillary pressure term (entry or breakthrough)

-Line 110: Shielding capacity -> sealing capacity?

-Line 116: observed using a point-counter

-Line 120: What software was used to identify the XRD peaks? Were they automatically picked or by hand?

-Figure 1: The topmost conglomerate core looks as if it is fractured in the middle.

-Line 125-126: Reads like a results section and not the experimental set-up.

-Line 136: Can you elaborate on the reasons for polishing the surfaces?

-Figure 2: Heating of the cell is not illustrated and should at least be mentioned in the caption.

-Line 160: Why use destilled water if the reservoir conditions are illustrated?

-Line 171: How long was the experiment?

-Line 179: in the Formula there are to minus signs?

-Line 187 & 190:Here f should be used for the Co2 storage ratio as this is the letter which has been introduced in line 179.

-Line 210: How long was it dried for?

-Line 210-225: Here the terms capillary entry pressure and breakthough pressure should be used.

-Line 230: The use of destilled water instead of brine should be explained.

-Line 258-262: Is copy and paste from methods and is not needed here.

Author Response

For Reviewer 3:

In their proposed article the authors study the important question of how much CO2 could be stored safely with in the the Janggi Basin, South Korea. They use laboratory experiments to evaluate the storage capacity for CO2 storage of sandstones and conglomeratic formations as well as experiments which analyse the sealing potential of the caprocks occurring in the basin. After discussing the experimental results, the authors conclude, that the conglomerates in the basin are an ideal storage formation and that enough storage space for a test injection site.

 While the manuscript covers an important part of any CO2 storage operation – identifying the amount of CO2 that can be safely stored within the reservoir – I find that several key issues need to be addressed before I can recommend publication. These are:

Question (1): The geological setting of the basin is not very well described. A stratigraphic column is a must, and if possible a cross section. At the moment it is not clear how the subsurface is structured. Are there any faults? The basin is very small (10 km2), how would this affect the pressure distribution during injection? Does the pore-water have space to escape to? How are the edges of the basin?

Answer (1): As the reviewers’ suggestion, the stratigraphic columnar section of the drilling site was added in Figure 1. (See Figure 1)

Several geological and structural studies for the subsurface of the potential CO2 storage testing site at the Janggi Basin have been done and they presented the depositional history of the Janggi Basin conglomerate in the Early Miocene in detail and gave us the structural and spatial domain of possible CO2 storage site in the Janggi Basin. Of course, the total domain of the Janggi Basin is much larger than 10 km2. The sentence in manuscript means that the secured volume to store safely CO2 is at least about 10 km2 based on the drilling data and recent previous geological researches (excluding possible leakage zones such as fault zones and the deficient layer thickness. Etc…). It is sure that if more data about the subsurface structure of the basin are acquired, the available CO2 storage volume in the Janggi Basin may increase. Because this paper focus on the laboratory works to evaluate the storage and the sealing capacity, the detail structural and geological information for the Janggi Basin in field scale exceed its scope. However, according to reviewer’s suggestion, several references dealing with geological and structural information for the Janggi Basin were added in manuscript and sentences were revised to describe more clearly the CO2 storage volume. The brief description of the Janggi Basin structure was also added for readers’ better understanding. (Line 44 -54). Here are reference lists added in the manuscript.

8. Gu, H.-C.; Hwang, I.G. Depositional history of the Janggi Conglomerate controlled by tectonic subsidence, during the early stage of Janggi Basin evolution. J. Geol. Soc. Korea 2017, 53, 221–240, doi.org/10.14770/jgsk.2017.53.2.221.

9. Gu, H.-C.; Gim, J.-H.; Hwang, I.G. Variation in depositional environments controlled by tectonics and volcanic activities in the lower part of the Seongdongri Formation, Janggi Basin. J. Geol. Soc. Korea 2018, 54, 21–46, http://dx.doi.org/10.14770/jgsk.2018.54.1.21.

Question (2): The authors use the term seepage pressure several to describe how CO2 migrates into or through caprocks. A much better term would be capillary entry pressure and capillary breakthrough pressure – these are widely used in the CO2 storage community (e.g. Iglauer, 2018, https://doi.org/10.1016/j.ijggc.2018.07.009, or Miocic et al., 2019, https://doi.org/10.5194/se-10-951-2019) and are best used to differentiate between the different pressures.

Answer (2): Thanks for the good suggestion. As the reviewer’s suggestion, the term “seepage pressure” was replaced by “the capillary entry pressure” in the whole manuscript and illustrations. One of them was added in the reference list (See ref. [24]).

Question (3) The discussion is lacking several important issues:

Question 3a) In the experiments distilled water was used, but other experiments have highlighted that the salinity of brine has a big impact on the capillary storage capability for CO2.

Answer 3a): Authors agreed with the reviewer to use real groundwater or brine water from the drilling site than distilled water. The real groundwater directly sampled from the drilling site or the artificial water similar to groundwater should be used when researches focus on the mineralogical or geochemical reactions at the storage rock. We already used genuine groundwater in the study on the scCO2-rock-groundwater reaction and several cited papers in this manucript were listed below.

26.   Park, J.; Baek, K.; Lee, M.; Wang, S. Physical property changes of sandstones in Korea derived from the supercritical CO2‒sandstone‒groundwater geochemical reaction under CO2 sequestration condition. Geosci. J. 2015, 19, 313–324, doi:10.1007/s12303-014-0036-4.

28.   Park, J.; Baek, K.; Lee, M.; Chung, C.-W.; Wang, S. The Use of the Surface Roughness Value to Quantify the Extent of Supercritical CO2 Involved Geochemical Reaction at a CO2 Sequestration Site. Appl. Sci. 2017, 7, 572, doi:10.3390/app7060572.

Because this study is dealing with the direct measurement of the scCO2 replaced in pores for 30 - 50 days, we thought groundwater quality and their effects on the property change of the rock can be ignored in this study. Also, most of groundwater directly sampled from the drilling site was consumed or contaminated in previous experiments and we have to use distilled water in this study. As reviewers’ suggestion, we will use the artificial groundwater similar to the real groundwater or real groundwater (if possible) for the further study. Authors earnestly ask for the reviewer’s understanding.     

Question 3b) It is unclear if CO2 dissolution into the pore water and residually trapped CO2 was considered when the storage capacity was calculated. During the experiments some volume of CO2 must have been residually trapped? In the introduction it is explained that the dissolved CO2 is neglected, as the direct storage is seen as “the substantive amount of CO2”. How much is that exactly?

Answer 3b): The provisional estimation of dissolved CO2 amount may be possible by using the solubility of CO2 in water at specific temp and pressure condition (here 50 oC and 100 bar). However, the solubility of CO2 is dependent on not only temp. and pressure but also many water quality parameters. Also, the direct measurement of dissolved CO2 mass in the cell at high temp. and pressure condition is not easy. As mentioned in Introduction, the Janggi Basin was selected as a feasibility testing site to store at the least 100,000 metric tons of CO2 before the large scale injection (more than 1,000,000 metric tons) in Korean peninsula. As the testing site, it is important that the CO2 storage capacity of the Janggi basin is larger than 100,000 metric tons at least and our results support reasonable data about it. Of course, the substantive CO2 storage amount in the storage rock increases if the dissolved CO2 amount is considered in this study. We think it is enough to measure the scCO2 amount replaced in pores of storage rock for the evaluation of the storage capacity in the Janggi Basin because the total CO2 storage amount (scCO2 + dissolved CO2) is always larger than only scCO2 amount. Thus, if the enough scCO2 amount is secured in pore spaces in this experiment, the rock can be evaluated as the positive storage site regardless of dissolved CO2 amount. However, as the reviewer’s suggestion, authors added the sentence to describe this limitation frankly in “Results and discussion” and “Conclusions” sections. (Line 309-314; Line 372-374)

Question (4) The description of the methods is not consistent, for some experimental steps the time specifications are not mentioned. The reservoir rocks consist of many rock fragments, fine matrix, clays and feldspars – while the authors have not observed much of an alteration during the 90 days of experiment, this amount of reactive minerals could be an issue during long-term (10.000 years) storage? A discussion on this is needed.

Answer (4): The times for several experimental steps were added in the manuscript (total time to measure replace scCO2, time in oven dried, total time to measure the capillary entry pressure. Etc..). (Line 192-193; 240; 249; 258)

Author completely agreed with the reviewer’s opinion about the physical or chemical property changes of rock, originated from the mineralogical or geochemical reaction with CO2. Our research group studied on the scCO2-sandstone-water reaction and investigated that this reaction occurs very active and affects the physical property changes within a short time than expected. See the paper of ① Park et al. (2015) Physical property changes of sandstones in Korea derived from the supercritical CO2-sandstone-groundwater geochemical reaction under CO2 sequestration condition, Geosciences Journal, 19, p. 313-324 and ② Lee et al. (2015) Investigation of the relationship between CO2 reservoir rock property change and the surface roughness change originating from the supercritical CO2-sandstone-groundwater geochemical reaction at CO2 sequestration condition, Energy Procedia, 76, 495-502. Authors agreed that the effect of the mineralogical and geochemical reaction on the storage capacity and on the leakage safety is very important but it is out of scope of this paper (It should be discussed as a different subject.). However, as the reviewer’s suggestion, the limitation of this paper and frank discussion for effect of the geochemical reaction were added at the end in “Conclusion” section. (Line 382-389)

Question (5)   Most of the paper is written well, however, the abstract needs some significant rewriting with regards to the English language.

Answer (5): Author(s) request the English proofreader to review the abstract in English before the resubmission.

Question (6) Some more minor issues:

- Line 35-36: Reference to figure 1 should be places here

Answer) Line 36-39: The reference was placed in the end of the sentence as reviewers’ suggestion (see reference [7]). Two new references were also added in the manuscript related to the structural information for the Janggi basin (See reference [8, 9]).

7. Kim, M.-C.; Gihm, Y.S.; Son, E.-Y.; Son, M.; Hwang, I.G.; Shinn, Y.J.; Choi, H. Assessment of the potential for geological storage of CO2 based on its structural and sedimentologic characteristics in the Miocene Janggi Basin, SE Korea. J. Geol. Soc. Korea 2015, 51, 253–271, doi:10.14770/jgsk.2015.51.3.253.

8. Gu, H.-C.; Gim, J.-H.; Hwang, I.G. Variation in depositional environments controlled by tectonics and volcanic activities in the lower part of the Seongdongri Formation, Janggi Basin. J. Geol. Soc. Korea 2018, 54, 21–46, http://dx.doi.org/10.14770/jgsk.2018.54.1.21.

9. Gu, H.-C.; Hwang, I.G. Depositional history of the Janggi Conglomerate controlled by tectonic subsidence, during the early stage of Janggi Basin evolution. J. Geol. Soc. Korea 2017, 53, 221–240, doi.org/10.14770/jgsk.2017.53.2.221.

- Line 44-45: Here it should be mentioned how the basin is separated from neighbouring areas.

Answer) The detail tectonic histories of the Janggi conglomerate formations exceed the scope of this paper. However, as the reviewers’ suggestion, authors added a paragraph for the description of the Janggi conglomerate formation and also added its reference which deals with the interpretation of the subsurface tectonic history and the spatial domain of the Janggi basin in detail (Line 44-54 and ref. [7,8,9]).

-Line 76: m depth

Answer) Authors changed words to “m depth” (Line 84).

-Line 80-82: For me duration of sequestration means the time of injection, but the storage site should be secure for the whole lifespan of at least 10.000 years.

Answer) In this paper, the sealing capacity of the cap rock means the hydrodynamical intrusion of the stored scCO2 in the reservoir into the upper part passing through the cap rock. The diffusion or the seepage originated from the geochemical reaction were not considered in this study. Authors agree that the upward movement of scCO2 may occur by the property change of the cap rock, resulting from any other parameters such as geochemical reaction for long periods of time. Description for effect of the geochemical reaction were added at the end in “Conclusion” section. (Line 382-389)

-Line 84 and onwards: Replace initial seepage pressure with the correct capillary pressure term (entry or breakthrough)

Answer) As the reviewer’s suggestion, the term “seepage pressure” was replaced by “the capillary entry pressure” in the whole manuscript and illustrations.

-Line 110: Shielding capacity -> sealing capacity?

Answer) Authors changed words to “sealing capacity” (Line 135).

-Line 116: observed using a point-counter

Answer) Authors changed words to “a point-counter” (Line 142).

-Line 120: What software was used to identify the XRD peaks? Were they automatically picked or by hand?

Answer) The phase determination for the XRD peaks was performed by using the software program of “PANalytical X’Pert HighScore Plus 2.0”.

-Figure 1: The topmost conglomerate core looks as if it is fractured in the middle.

Answer) The photograph of the conglomerate core was its original state taken from the drilling cores. Of course, the part without cracks or fractures of the core was used in experiments.

-Line 125-126: Reads like a results section and not the experimental set-up.

Answer) The first sentence in section 2.2 explains why the experiment is necessary in manuscript and it is closely related to the definition of the scCO2 storage ratio. Authors think it will be helpful for the readers’ better understanding. Authors ask for the reviewer’s understanding for the sentence.

-Line 136: Can you elaborate on the reasons for polishing the surfaces?

Answer) It is because the more flat surface results in the more constant pressure or the more uniform flow intrusion on the cut surface of the core. As the reviewers’ suggestion, authors added the description for it in the sentence. (Line 162-164)

“The sandstone and conglomerate cores were cut (4.2 cm in diameter and 5–7 cm in length) and their cut surfaces were polished using powdered diamond paper to maintain a uniform scCO2 or water injection pressure on the cut surface.”

-Figure 2: Heating of the cell is not illustrated and should at least be mentioned in the caption.

Answer) The heating of the cell was operated by the heating jacket and it was mentioned in the sentence (Line 184-185). In Figure 3, the heating jackets used in the experiment were also illustrated in red color. (see Figure 3)

-Line 160: Why use destilled water if the reservoir conditions are illustrated?

Answer) Authors already answered this question in Question (3(a)) in detail. Please, see the answer (3(a)).

-Line 171: How long was the experiment?

Answer) The measurement of the scCO2 storage ratio for the conglomerate and sandstone cores was conducted after at least two pore volumes of scCO2 were flushed from the core. Thus, the flushing time for water and scCO2 are different according to the porosity and the flow velocity in pores of rock core. But the flushing time for rock core was less than 7 days (mostly within a few days and faster in conglomerates). Total time for the scCO2 storage measurement experiment was around 3-4 weeks for each core. As reviewers’ suggestion, author added the scCO2 flushing time in the sentence. (Line 193; 249)

-Line 179: in the Formula there are to minus signs?

-Line 187 & 190:Here ε should be used for the Co2 storage ratio as this is the letter which has been introduced in line 179.

Answer) Thanks for both of good suggestion. As the reviewers’ suggestion, equation (1) and (2) were changed. (Please see the revised manuscript)

-Line 210: How long was it dried for?

Answer) Depending on the moisture content in each core. The total time to dry each core in this experiment was 7 days and authors added the drying time as 7 days in the sentence as reviewers’ suggestion. (Line 240)

-Line 210-225: Here the terms capillary entry pressure and breakthough pressure should be used.

Answer) As the reviewer’s suggestion, the term “seepage pressure” was replaced by “the capillary entry pressure” in the whole manuscript and illustrations.

-Line 230: The use of distilled water instead of brine should be explained.

Answer) Authors already answered this question in Question (3(a)) in detail. Please, see the answer (3(a)).

-Line 258-262: Is copy and paste from methods and is not needed here

Answer) As reviewers’ suggestion, the sentences were removed here and were located in section 2.2 (the method section) (Line 189 - 193)

--------------------------------------------------------- The end ----------------------------------------------------

Round 2

Reviewer 2 Report

The Authors referred to the results of their previous research, and highlighted the original achievements presented in the reviewed version of the paper.

Style of the text still needs improvement to meet the requirements of scientific journals.

Conclusion: the paper was corrected to the extent, that allows its publication, after some linguistic amendments, in the MDPI Minerals Journal.

Author Response

Dear reviewer B;

Please find an enclosed file of the second revised manuscript. Responding to call from the reviewer B, the paper was proofread in English again by a native proofreader to improve the English style with linguistic amendments. The corrected parts were highlighted in red color and they are easily visible to reviewers.

Thank you for your review again.
